

# Integrative analysis of the transcriptome and metabolome provides insights into polysaccharide accumulation in *Polygonatum odoratum* (Mill.) Druce rhizome

Gen Pan[1,2,3], Jian Jin[1], Hao Liu[1], Can Zhong[1], Jing Xie[1], Yuhui Qin[2] and Shuihan Zhang[1]

[1] Institute of Chinese Medicine Resources, Hunan Academy of Chinese Medicine, Changsha, Hunan, China
[2] Colleges of Chinese Medicine, Hunan University of Chinese Medicine, Changsha, Hunan, China
[3] Institute of Bast Fiber Crops, Chinese Academy of Agricultural Sciences, Changsha, Hunan, China

## ABSTRACT

**Background**. *Polygonatum odoratum* (Mill.) Druce is a traditional Chinese herb that is widely cultivated in China. Polysaccharides are the major bioactive components in rhizome of *P. odoratum* and have many important biological functions.

**Methods**. To better understand the regulatory mechanisms of polysaccharide accumulation in *P. odoratum* rhizomes, the rhizomes of two *P. odoratum* cultivars 'Y10' and 'Y11' with distinct differences in polysaccharide content were used for transcriptome and metabolome analyses, and the differentially expressed genes (DEGs) and differentially accumulated metabolites (DAMs) were identified.

**Results**. A total of 14,194 differentially expressed genes (DEGs) were identified, of which 6,689 DEGs were down-regulated in 'Y10' compared with those in 'Y11'. KEGG enrichment analysis of the down-regulated DEGs revealed a significant enrichment of 'starch and sucrose metabolism', and 'amino sugar and nucleotide sugar metabolism'. Meanwhile, 80 differentially accumulated metabolites (DAMs) were detected, of which 52 were significantly up-regulated in 'Y11' compared to those in 'Y10'. The up-regulated DAMs were significantly enriched in 'tropane, piperidine and pyridine alkaloid biosynthesis', 'pentose phosphate pathway' and 'ABC transporters'. The integrated metabolomic and transcriptomic analysis have revealed that four DAMs, glucose, beta-D-fructose 6-phosphate, maltose and 3-beta-D-galactosyl-sn-glycerol were significantly enriched for polysaccharide accumulation, which may be regulated by 17 DEGs, including UTP-glucose-1-phosphate uridylyltransferase (*UGP2*), hexokinase (*HK*), sucrose synthase (*SUS*), and UDP-glucose 6-dehydrogenase (*UGDH*). Furthermore, 8 DEGs (*sacA*, *HK*, *scrK*, *GPI*) were identified as candidate genes for the accumulation of glucose and beta-D-fructose 6-phosphate in the proposed polysaccharide biosynthetic pathways, and these two metabolites were significantly associated with the expression levels of 13 transcription factors including *C3H*, *FAR1*, *bHLH* and *ERF*. This study provided comprehensive information on polysaccharide accumulation and laid the foundation for elucidating the molecular mechanisms of medicinal quality formation in *P. odoratum* rhizomes.

Corresponding authors
Yuhui Qin, 1243695133@qq.com
Shuihan Zhang, zhangshuihan0220@126.com

## INTRODUCTION

*Polygonatum odoratum* (Mill.) Druce (*P. odoratum*), a traditional Chinese medicine, is a perennial herbaceous plant that is widely distributed in East Asia and Europe (*Zhao et al., 2018*). In China, *P. odoratum* is a traditional and bulk Chinese herbal medicine, and the medicinal part of the plant is the rhizome, which has been extensively used to treat certain diseases, such as heart disease, diabetes, and tuberculosis (*Yu, 1993*). The physiological function of *P. odoratum* is mainly due to its rhizome, which is rich in various bioactive substances, including polysaccharides, flavonoids, dipeptides, and mineral elements (*Chen et al., 2014*). Among these active components, high contents of polysaccharides are the markers of the medicinal quality of *P. odoratum* and exert its pharmacological activity (*Jiang et al., 2013*).

Polysaccharides, defined as polymer carbohydrates comprising at least ten monosaccharides linked by glycosidic bonds (*Zeng et al., 2019*), play crucial roles in various physiological processes. In *P. odoratum*, they form a heteropolysaccharide composed of mannose, galactose, glucose, arabinose, and galacturonic acid, imparting diverse pharmacological properties such as blood glucose reduction, immune regulation, anti-tumor activity, antioxidant effects, anti-fatigue properties, and delayed skin aging (*Deng et al., 2012*; *Zhao et al., 2019a*; *Zhao et al., 2019b*). Studies have discovered that polysaccharides biosynthesis in plants includes three main steps (*Zhang et al., 2020*; *Chen et al., 2022*). During the first step, sucrose undergoes a series of transformations to produce uridine diphosphate (UDP)-glucose, guanosine diphosphate (GDP)-mannose, and guanosine diphosphate-fucose (*Richez et al., 2007*; *Decker et al., 2012*). The second step is the conversion of UDP-glucose to its nucleotide-diphospho (NDP) monosaccharide (*Yin et al., 2011*; *Li et al., 2022*). Finally, different glycosyltransferases (GTs) remove monosaccharides from the sugar nucleotides. The donor binds to the growing polysaccharide polymer, and these repeating units are then polymerized and exported to form plant polysaccharides (*Lairson et al., 2008*; *Pauly et al., 2013*). Previous studies have revealed that polysaccharides biosynthesis requires continuous enzymatic reactions related to certain enzymes, including invertase (*INV*/*sacA*), hexokinase (HK), GT, and fructokinase (*scrK*) (*Kadokawa, 2011*). Sucrose is converted into glucose (Glc) and fructose (Fru) by *INV*/*sacA*. HK and *scrK* convert Glc and Fru into glucose 6-phosphate (Glc-6P) and fructose 6-phosphate (Fru-6P), respectively. Subsequently, they are used to synthesize UDP-glucose (UDP-Glc) and guanosine diphosphomanose (GDP-Man). Finally, UDP-Glc and GDP-Man are synthesised *via* GT reactions (*Kadokawa, 2011*).

The genes involved in polysaccharides biosynthesis have been extensively studied in cereal crops, including rice, corn, and wheat (*Huang et al., 2021*; *Li, Tan & Zhang, 2021*). A series of key enzymes, such as ADP-glucose pyrophosphorylase (AGPase), granule-bound starch synthase (GBSS), soluble starch synthase (SS), starch branching enzyme (SBE),

debranching enzyme (DBE), disproportionating enzyme (DPE), and starch/a-glucan phosphorylase (PHO), have been shown to participate in the synthesis of starch in cereal crops (*Li et al., 2021*). Certain genes that encode key enzymes have been characterized as significant genes involved in starch biosynthesis. For example, *AGPS* and *AGPL*, which encode AGPases, are responsible for synthesising ADP glucose (ADPG), the major substrate for starch synthesis in rice (*Prathap & Tyagi, 2020*). *OsGBP*, which encodes a protein that targets starch, is involved in the starch synthesis pathway through its interaction with granule-bound starch synthase I (GBSSI) in rice (*Wang et al., 2020*). In Chinese herbs, *DoCSLA6* and *DoGMT* are critical for mannan polysaccharides in *Dendrobium officinale*, and *DoCSLA6* overexpression can enhance their content in *A. thaliana* (*He et al., 2017*; *Yu et al., 2018*). Although candidate genes possible related to polysaccharide synthesis in *P. odoratum* were studied based on comparative transcriptomic analysis of different tissues, no studies have applied multi-omics approaches to explore the pathway, crucial metabolism and key genes governing polysaccharides accumulation in the *P. odoratum* rhizome, and its mechanism in rhizome remains unknown.

Transcriptomic and metabolomic analyses have emerged as efficient methods for exploring complex biochemical pathways and identifying key genes and metabolites involved in the biosynthesis of bioactive compounds in plants (*Park et al., 2021*; *Yu et al., 2022*; *Wang et al., 2023a*; *Wang et al., 2023b*). Previous studies have identified the key genes associated with the bioactive metabolite synthesis pathways of plants, such as *Dendrobium officinale*, *Cryptomeria fortunei*, *Cordyceps militaris*, and *Tetrastigma hemsleyanum*, using transcriptome and metabolome analyses (*Yuan et al., 2022*; (*Zhang et al., 2022*; *Wang et al., 2023a*; *Wang et al., 2023b*; *Hang et al., 2023*). Through the transcriptome and metabolome analyses, nine genes (*HCT, CHS, CHI, F3H, F3'H, F3'5'H, FLS, DFR, and LAR*) have been identified as the key genes for flavonoid biosynthesis in *Cryptomeria fortunei* (*Zhang et al., 2022*). In *Dendrobium officinale*, transcriptome and metabolome profiling have revealed that three key *FLS* genes regulate flavonoid accumulation (*Yuan et al., 2022*). These findings have indicated that a transcriptome analysis combined with a metabolome analysis may be an effective method for identifying key genes involved in the biosynthesis of bioactive substances. However, such comprehensive studies identified the key gene networks controlling polysaccharides accumulation in *P. odoratum*. are lacking.

The rhizome is the main component for medicinal use in *P. odoratum* because it possesses many medicinal properties, mainly composed of polysaccharides. Thus, in this study, the rhizomes of two varieties of *P. odoratum* with different polysaccharide contents were used for transcriptome and metabolome analyses. We further focused on the possible key genes and crucial metabolites for polysaccharide accumulation by integrating multi-omics studies. Our findings could provide theoretical support for analysis of polysaccharide accumulation and elucidating the mechanisms underlying medicinal quality formation.

## MATERIALS AND METHODS

### Plant materials

Three *P. odoratum* plants from 11 cultivars were collected from Wangcheng Base (Institute of Chinese Medicine Resources; Hunan Academy of Chinese Medicine) on August 18, 2022. All the cultivars were identified by Associate Professor Gen. Pan at the Institute of Chinese Medicine Resources; Hunan Academy of Chinese Medicine. After washing with sterile distilled water, the rhizomes were immediately frozen in liquid nitrogen and stored at −80 °C for RNA sequencing (three biological samples for two cultivars), metabolomic analysis (six biological samples for two cultivars) and total RNA extraction.

### Extraction and determination

Eleven rhizomes of different *P. odoratum* cultivars were dried at 80 °C to a constant weight and then ground into powder to determine polysaccharides. The total polysaccharides content were detected using the phenol-sulfuric acid method described by the (*Chinese Pharmacopoeia Commission, 2020*). Determining polysaccharides content from each cultivar included three biological and three technical replicates.

### Total RNA extraction, cDNA library construction, sequencing, and analyses

The total RNA from five tissues (rhizomes, stems, leaves, roots, and leaves) of *P. odoratum* and rhizomes of four genotypes ('Y3', 'Y6', 'Y10' and 'Y11') was extracted using an EASYspin Plus Plant RNA Kit (Aidlab Biotechnologies Co., Ltd., Beijing, China). The quality and quantity of the total RNA were assessed on agarose gels and determined using a NanoDrop 2000 instrument (Thermo Fisher Scientific, Waltham, MA, USA), and then 1 µg of total RNA of the rhizome of two cultivars was prepared for cDNA synthesis using an NEBNext® Ultra™ II RNA Library Prep Kit (New England Biolabs Inc., Ipswich, MA, USA) according to the protocol. The quantity and purity of the library were checked using a Qubit 2.0 Fluorometer (Life Technologies, CA, USA) and a Bioanalyzer 2100 system (Agilent Technologies, Santa Clara, CA, USA). Finally, the RNA sequencing was performed using an Illumina NovaSeq 6000 platform (Illumina) by Shanghai OE Biotech Co., Ltd. (Shanghai, China).

After processing and filtering the raw data using Trimmomatic to remove reads containing poly-N and low-quality reads, clean data were obtained. The fragments per kilobase of the exon model per million mapped fragments (FPKM) value for each unigene was calculated using Cufflinks. The differentially expressed genes (DEGs) with $P < 0.05$ and foldchange (FC)>2 or <0.5 were identified using DESeq. The Gene Ontology (GO) enrichment and Kyoto Encyclopedia of Genes and Genomes (KEGG) pathway enrichment analyses were performed using all the DEGs.

### Identification of transcription factors (TF)

The open reading frames (ORFs) of the unigenes were examined and were then aligned to the protein domain of the transcription factor using hmmsearch (http://hmmer.org). The unigenes were annotated using the PlantTFDB (Plant transcription factor database). By

comparison with Pfam23.0, the unigenes encoding the transcription factors were further identified.

### Rhizome metabolites liquid chromatography-mass spectrometry analysis of rhizome metabolites

The metabolites were extracted from the rhizomes of the two *P. odoratum* cultivars (approximately 60 mg) using methanol. After extraction, the metabolites were measured using a Dionex Ultimate 3000 RS UHPLC system coupled with a Q-Exactive quadrupole-Orbitrap mass spectrometer (MS) (Thermo Fisher Scientific, Bremen, Germany) at Shanghai Lu-Ming Biotech Co., Ltd. (Shanghai, China).

### Untargeted metabolomics data

The obtained GC/MS raw data were analysed using Progenesis QI software (Waters Corporation, Milford, United States) based on four databases: the public databases Human Metabolome Database (HMDB), Lipidmaps (v2.3), and METLIN and a self-built database (Luming Biotech Co., Ltd., Shanghai, China). The differentially accumulated metabolites (DAMs) were identified using the parameters VIP > 1 and $p < 0.05$. All the DAMs were used for the KEGG pathway enrichment analyses.

### Real-time quantitative reverse transcription PCR (RT-qPCR) analysis

The rhizomes of 'Y11' and 'Y10' were used to validate the RNA-seq data, and five different tissues, including the root, leaf, rhizome, stem, and flower, were collected from 'Y11' and used to tissue-specific expression patterns analysis. The reverse transcriptions were performed according to the instructions of the Evo M-MLV RT Premix for qPCR Kit (Accurate, Changsha, China), and the SYBR® Green Premix Pro Taq HS qPCR Kit (Accurate, Changsha, China) was used for the real-time quantitative reverse transcription PCR (qRT-PCR). The *actin* gene was used as an housekeeping gene as previously reported (*Zhang et al., 2020*). The primers used for the qRT-PCR analysis are listed in Table S2. The experiment was repeated with at least two biological replicates. The data analysis was performed using the $2^{-\Delta\Delta Ct}$ method.

### Statistical analysis

One-way analysis of variance (ANOVA) was performed using IBM SPSS Statistics version 20 software (SPSS Inc., Chicago, IL). Values were mean values and SD of at least two independent experiments with three replicates for each. The correlation coefficients between the content of DAMs and the gene expression levels of DEGs were analysed using Pearson's correlation coefficient and Student's $t$-test.

## RESULTS

### Determination of rhizome polysaccharide content in eleven *P. odoratum* cultivars

Eleven *P. odoratum* cultivars were used to select the two cultivars with the most distinct total polysaccharide contents for the transcriptome and metabolic analyses. As shown in Fig. S1, the polysaccharides contents of the 11 materials ranged from 1.39% to 6.87%.

(A)

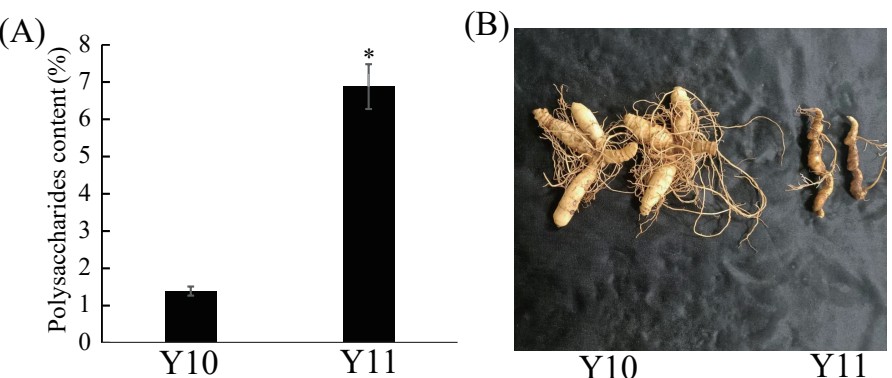

(B)

**Figure 1** **Polysaccharides content (A) and representative images (B) of rhizome in two *P. odoratum* cultivars 'Y10' and 'Y11'.** All date are means (±SD), *n* = 3. Significant differences were determined using one-way ANOVA: * *P* < 0.05.

Among these, the polysaccharide content of the 'Y10' rhizome was the lowest, whereas that of the Y11 rhizome was the highest (Fig. 1A). In addition, the two materials had significant differences in the rhizome shape and colour (Fig. 1B). Thus, these two cultivars were used for further study.

## Differentially expressed genes (DEGs) analysis

To analyse the differences in the transcriptome profiles associated with polysaccharide accumulation, the 'Y10' and 'Y11' rhizomes were used for an RNA-seq analysis. A total of 43 clean bases were obtained, and the clean reads varied from 46.51M to 50.44M, with an average Q30-value of 92.96 and GC content of 47.8 (Table S1). After *de novo* assembly and removing redundant transcripts, the transcriptome contained 71,432 unigenes. Comparing the gene expressions of 'Y10' with those in 'Y11', a total of 14,194 DEGs were obtained, of which 7,505 genes were upregulated and 6,689 genes were downregulated (Fig. S2). As described in the bubble diagram of the KEGG enrichment results, the top 20 pathway classifications of up-regulated DEGs were enriched in 14 metabolism pathways, which were mainly related to 'terpenoid backbone biosynthesis', ''alanine, aspartate and glutamate metabolism' and 'glycine, serine and threonine metabolism' (Fig. 2A). Meanwhile, the enrichments of all the downregulated DEGs were associated with 15 metabolism pathways, 'starch and sucrose metabolism' was the most abundant, followed by 'amino sugar and nucleotide sugar metabolism'. Next, a total of 15 pathways classifications were screened from the DEGs related to carbohydrate metabolism. Most of them are most likely involved in polysaccharide accumulation, such as 'glycolysis/gluconeogenesis', 'starch and sucrose metabolism', 'fructose and mannose metabolism', 'amino sugar and nucleotide sugar metabolism' and 'pentose and glucuronate interconversions'. Furthermore, in these pathways related to carbohydrate metabolism, the number of down-regulated DEGs was higher than that of up-regulated DEGs, with the exception of the 'Citrate cycle (TCA cycle)' pathway (Fig. 2B).

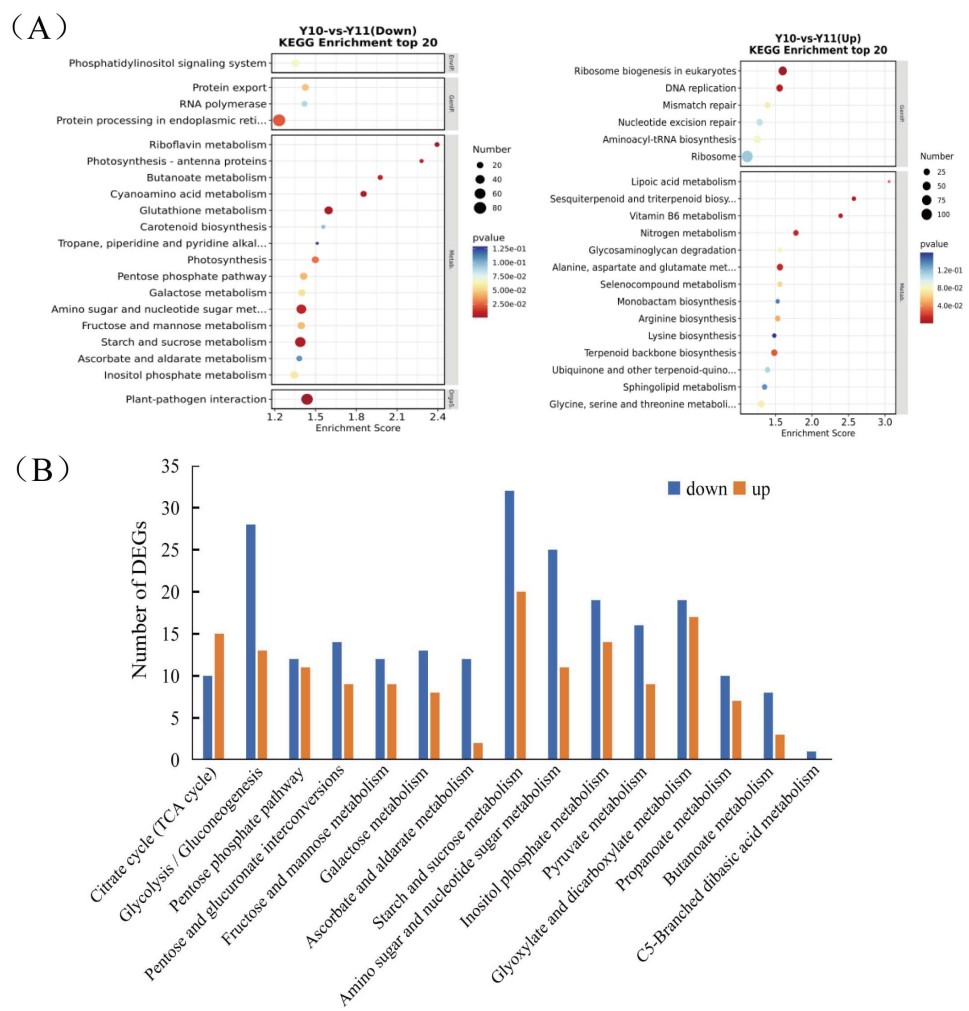

**Figure 2** **Functional enrichment analysis of differentially expressed genes (DEGs) by Kyoto Encyclopedia of Genes and Genomes (KEGG) pathway-based enrichment analysis.** (A) The top 20 pathway classifications selected by KEGG pathway-based enrichment analysis based on the DEGs between 'Y11' and 'Y10'. (B) The number of up-regulated or down-regulated DEGs related to the 15 pathway classifications for carbohydrate metabolism in the 'Y10' *vs* 'Y11' group.

## Metabolite profiles of rhizomes in 'Y11' and 'Y10'

To broadly evaluate the metabolite profile differences between 'Y11' and 'Y10', untargeted metabolomics of the rhizome were performed using UHPLC-QTOF-MS. A total of 312 metabolites with known structures were identified, of which 80 were differentially accumulated metabolites (DAMs) in 'Y10' compared to those in 'Y11'. Of the these DAMs, 52 were downregulated, whereas 28 DAMs were upregulated in 'Y10' compared to those in 'Y11' (Fig. 3). The KEGG enrichment analysis results indicated that the most abundant downregulated DAMs were enriched in 'tropane, piperidine and pyridine alkaloid biosynthesis', followed by 'ABC transporters' and 'pentose phosphate pathway' (Fig. 4A). In addition, the up-regulated DAMs enrichment of 'Y10' *vs* 'Y11' was most significant in 'ABC
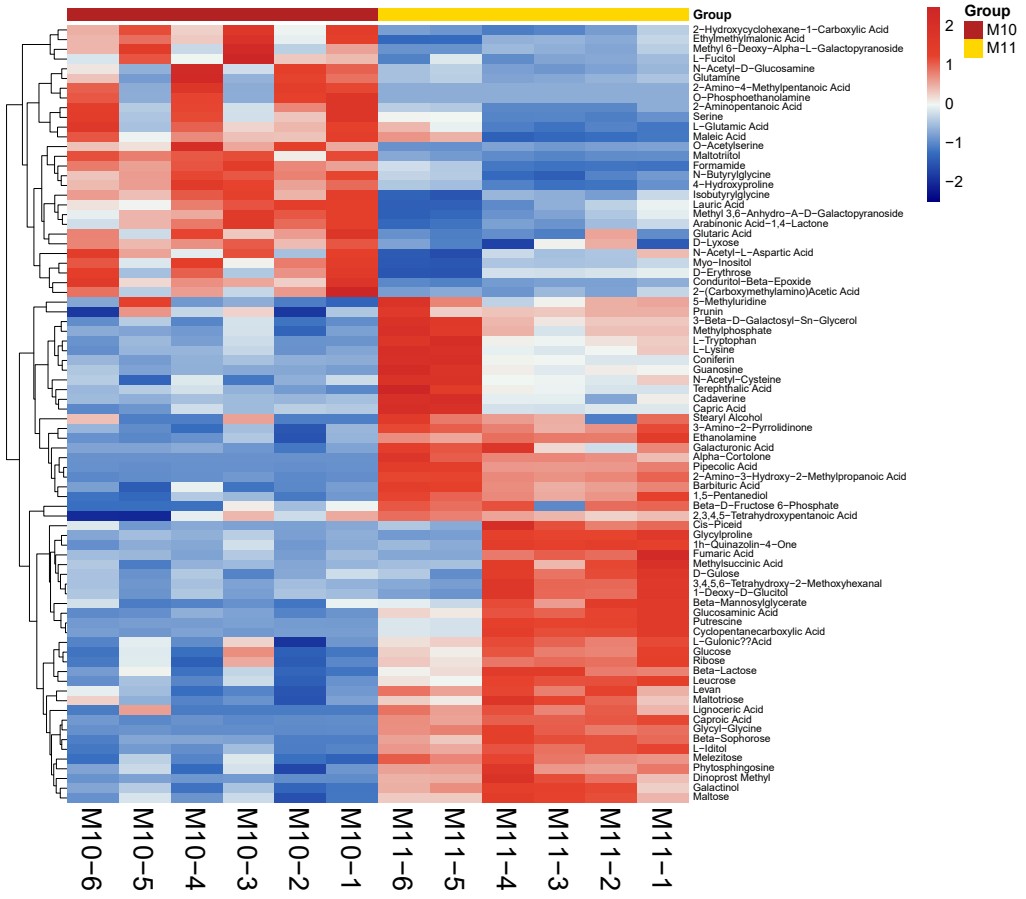

**Figure 3   Heatmap visualization of differentially accumulated metabolites (DAMs) of rhizome between 'Y11' and 'Y10'.** M10-1 M10-6: The six simples of 'Y10' for metabolome analysis; M11-1 M11-6: The six simples of 'Y11' for metabolome analysis.

transporters', 'nitrogen metabolism' and 'alanine, aspartate and glutamate metabolism'. Of these, the pathways closely associated with carbohydrate metabolism including 'the pentose phosphate pathway', 'glycolysis/gluconeogenesis', 'starch and sucrose metabolism', 'galactose metabolism', 'amino sugar and nucleotide sugar metabolism', and 'fructose and mannose metabolism' (Fig. 4A). Meanwhile, 12 DAMs, including seven down-regulated and five up-regulated DAMs, were enriched in 11 pathways associated with carbohydrate metabolism (Figs. 4B, 4C).

## Combined transcriptome and metabolome analyses

Combined transcriptomic and metabolomic analyses were performed to thoroughly investigate the changes in rizhomes of the two varieties for polysaccharide accumulation. According to the results of the KEGG enrichment analysis, the shared pathways of the DEGs and DAMs that were closely related to carbohydrate metabolism in the top 20 pathways were 'pentose phosphate pathway', 'amino sugar and nucleotide sugar metabolism', 'starch and sucrose metabolism', 'galactose metabolism', and 'fructose and mannose metabolism'.

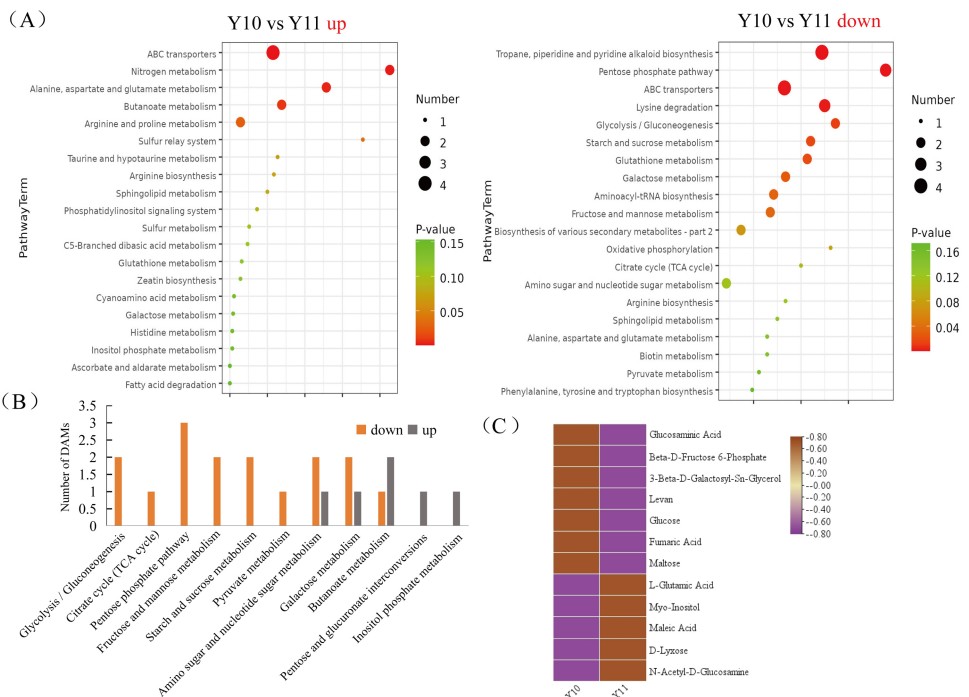

**Figure 4** **Functional enrichment analysis of differentially accumulated metabolites (DAMs) by Kyoto Encyclopedia of Genes and Genomes (KEGG) pathway-based enrichment analysis.** (A) The top 20 pathway classifications selected by KEGG pathway-based enrichment analysis based on the DAMs between 'Y11' and 'Y10' group. (B) The number of up- or down-regulated DAMs related to the 11 pathway classifications for carbohydrate metabolism in the 'Y10' *vs* 'Y11' group. (C) Heatmap visualisation of up-regulated and down-regulated DAMs related to the 11 pathway classifications for carbohydrate metabolism.

The pathways-genes-metabolites interaction network revealed that four DAMs, including glucose, maltose, 3-beta-D-galactosyl-sn-glycerol and beta-D-fructose 6-phosphate, were further observed in these commom pathways, and could be regulated by 17 DEGs (Fig. 5A). Meanwhile, the correlation analysis between these DEGs and DAMs were performed. As shown in Fig. 5B, three DEGs were positively correlated with beta-D-fructose 6-phosphate, one DEG was negatively correlated with glucose, three DEGs were negatively correlated with maltose while another three DEGs were negatively correlated with maltose, and four DEGs were negatively correlated with maltose while another three DEGs were negatively correlated with maltose (Fig. 5B). Among these 17 DEGs, the 12 DEGs, including *HK* and *UGP2*, were down-regulated in 'Y10' compared with those in 'Y11', while five DEGs, such as *SUS* and *UGDH*, were up-regulated (Fig. S4). As the four genes (*HK*, *UGP2*, *SUS* and *UGDH*) were the key enzyme genes reported to be involved in plant polysaccharide biosynthesis, their expression patterns were further analyzed in five tissues (root, leaf, flower, stem, and rhizome) and in the rhizomes of four genotypes with distinctly different polysaccharide contents. As shown in Fig. 6A, except for *UGDH* and *UGP2*, the other two genes were preferably expressed in the rhizomes, the main tissue for polysaccharide accumulation. Meanwhile, the expression levels of *SUS* and *HK* showed similar or opposite tendency to

Peer J

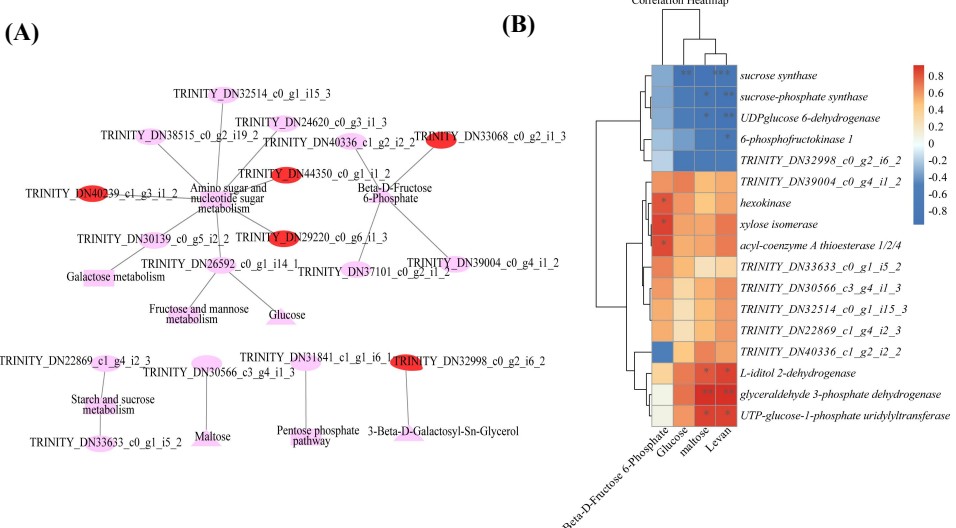

**Figure 5** Correlation analysis of DAMs and DEGs related to carbohydrate metabolism (A) Genes-metabolites interaction network. (B) Heat map of the correlation coefficients between four DAMs and 17 DEGs. Triangle: DAMs; Circle marked in red: the up-regulated genes in 'Y10' compared with those in 'Y11'; Circle marked in pink: the down-regulated genes in 'Y10' compared with those in 'Y11'. Asterisks represent statistical significance determined by Student's t-test (*P < 0.05, **P < 0.01).

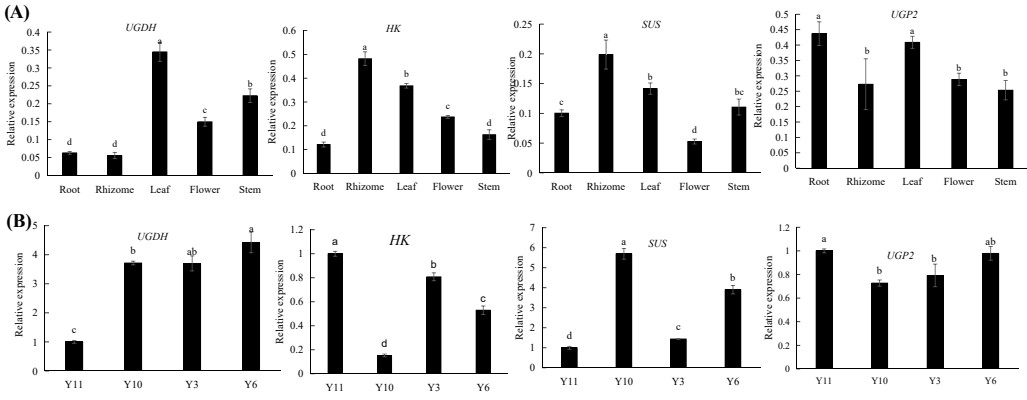

**Figure 6** (A–B) Expression pattern analysis of *SUS, HK, UGDH* and *UGP2* at different tissues and different genotypes. Different letters indicate significant difference at *P* < 0.05.

the polysaccharide content of four genotypes, and there was no obvious tendency between the transcript levels of *UGP2* and *UGDH* and the polysaccharide content in four genotypes (Fig. 6B).

## Identification of metabolites and genes related to polysaccharide biosynthesis

Based on the reported plant polysaccharide metabolic pathways, we further integrated our metabolomic and transcriptomic datasets and mapped the proposed polysaccharide
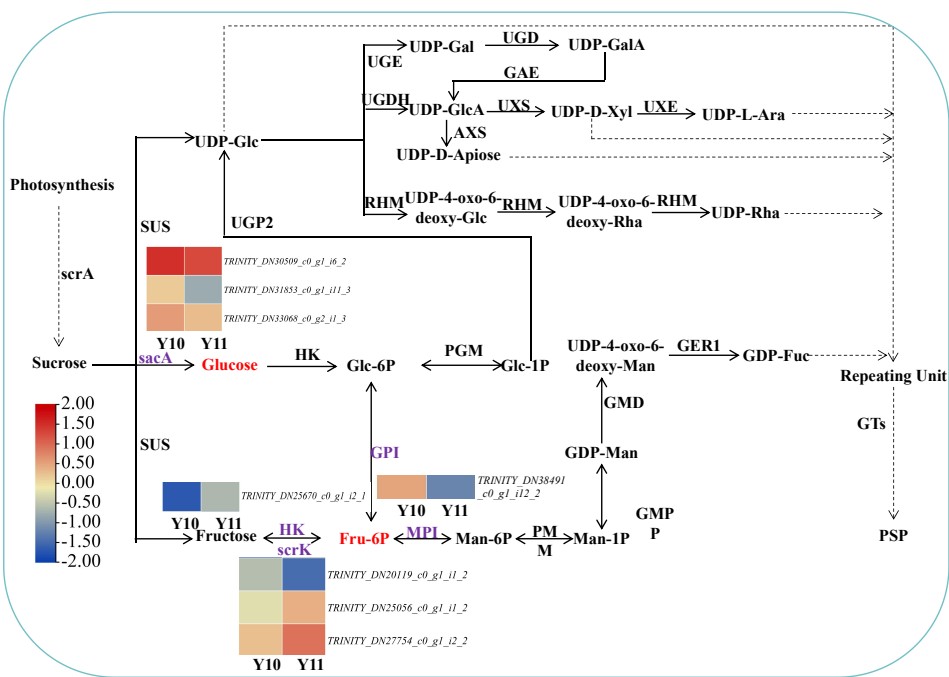

**Figure 7  Proposed biosynthetic pathways of polysaccharide in *P. odoratum* rhizome.** The up-regulated DAMs in 'Y11', compared with 'Y10', were marked in red; key enzymes encoded by nine DEGs related to the above-mentioned DAMs were marked in purple.

biosynthetic pathway of *P. odoratum* (Fig. 7). In this pathway map, it was observed that 'glucose' and 'beta-D-fructose 6-phosphate' accumulated more in 'Y11' than those in 'Y10'. Glucose is converted from sucrose by the enzyme beta-fructofuranosidase (INV/sacA), and beta-D-fructose 6-phosphate is converted from fructose, glucose-6-phosphate or mannose-6-phosphate by the enzymes hexokinase (HK), 6-phosphofructokinase (FRK/*scrK*), glucose-6-phosphate isomerase (GPI) and mannose-6-phosphate isomerase (MPI), respectively. In total, three DEGs annotated with *scrK* genes, three DEGs annotated with *sacA* genes, one DEGs annotated with *HK* gene, one DEGs annotated with *GPI* genes were identified. However, no one DEGs annotated with *MPI* gene were obtained. Among these eight DEGs, two *scrK* DEGs and one *HK* DEG showed higher expression levels in 'Y11' than those in 'Y10' (Fig. 7). These results suggest that the up-regulation of the three DEGs in Y11 may improve the efficiency of biosynthetic polysaccharide precursors in *P. odoratum*, which may account for the accumulation of polysaccharide in rhizomes.

## Identification of transcription factors involved in polysaccharide accumulation

In order to explore the transcriptional regulation of the genes involved in the accumulation of polysaccharide , we analysed the transcription factors (TFs) in the rhizomes. In total, we found that 229 TFs were significantly up/downregulated in Y10 rhizomes compared to those in Y11 rhizomes. Among these genes, 134 TFs were downregulated, while 95 TFs were upregulated. The most abundant transcription factor is AP/ERF-ERF (21) and

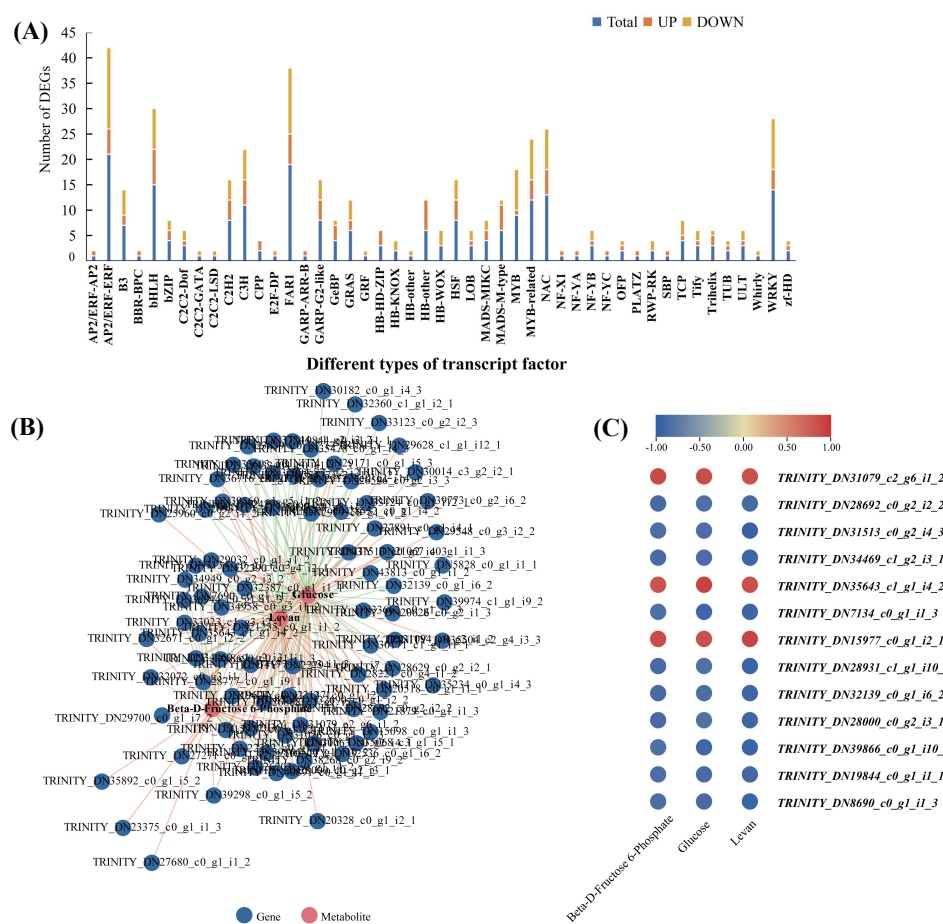

**Figure 8** **The transcription factors (TFs)-metabolite association network constructed according to the differentially expressed TFs and differentially accumulated metabolites involved in proposed pathways for polysaccharide biosynthesis in *P. odoratum*.** (A) Type and number of the differentially expressed TFs in *P. odoratum* rhizome. (B) TFs-metabolite association network analysis. The red lines indicate TFs that were positively significantly associated with the three metabolites, while the blue lines indicate TFs that were negatively significantly associated with the three metabolites. (C) Heatmap of the correlation coefficients between glucose, levan and beta-D-fructose 6-phosphate and the expression levels of TFs based on the Pearson's correlation coefficient.

MYB (21), followed by FAR1 (19) and WRKY (14) (Fig. 8A). Furthermore, correlation analysis revealed that 85 TFs were significantly associated with the metabolites glucose accumulation, 78 TFs with levan and 38 TFs with beta-D-fructose 6-phosphate content (Fig. S4). In addition, 13 TFs such as C3H, FAR1, bHLH and ERF were found to have high correlations with these three metabolites (Fig. 8B,C). These results suggest that polysaccharide accumulation may be regulated by these TFs .

## Verification of randomly selected DEGs using quantitative real-time PCR (qRT-PCR)

To verify the results of the RNA-seq analysis, nine DEGs were selected for the qRT-PCR analysis, of which four DEGs were related to 'galactose metabolism', 1 DEG related

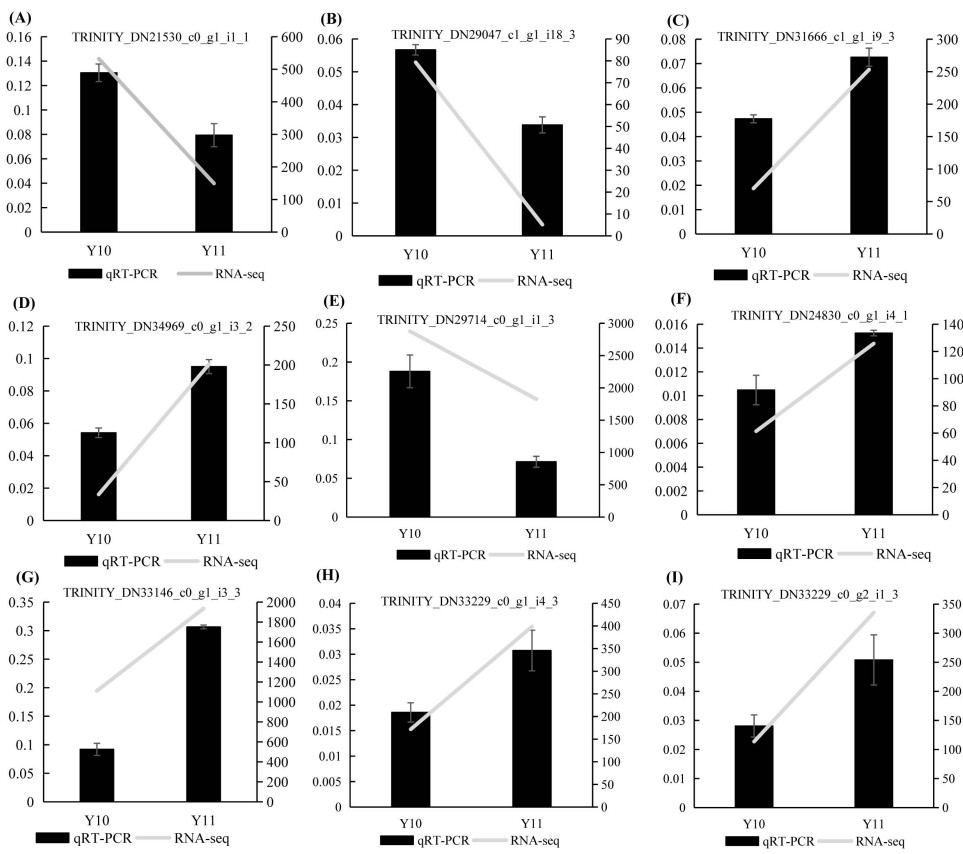

**Figure 9** (A–I) **The relative expression levels of nine selected DEGs were compared by RNA-seq and qRT PCR.** The line chart shows the gene expression level from the transcriptome (FPKM).

to 'glycolysis/gluconeogenesis', and four DEGs were related to 'fructose and mannose metabolism'. *Actin* was used as a reference gene. The results showed that the expression counts of the transcriptome data of the tested genes had a similar transcript profile to the qRT-PCR results (Fig. 9), indicating that the RNA-seq results obtained in this study were reliable and accurate.

## DISCUSSION

### First applications of multiomics to investigate the molecular mechanism of polysaccharide accumulation in *P. odoratum*

The rhizome is the main tissue for the medicinal use of *P. odoratum* and is rich in polysaccharides, a marker of medicinal quality. Although polysaccharides have several significant pharmacological functions, the molecular mechanism of polysaccharide biosynthesis remain largely unknown. To further explore the genes that encode the key enzymes that regulate polysaccharide biosynthesis, transcriptome and metabolome sequencing of the rhizomes of the two varieties with different polysaccharide contents was performed. In previous studies, only transcriptome analysis were performed to investigate the candidate genes involved in polysaccharide accumulation (*Wang et al.,*

*2017*; *Zhang et al., 2020*; *Li et al., 2022*). However, multiomics combined analysis has greater advantages than single transcriptome analysis in comprehensively explaining the regulatory mechanisms of the accumulation of active ingredients by associating the changes in metabolites with gene expression in medicinal plants (*Yuan et al., 2022*; *Zhang et al., 2022*). To the best of our knowledge, this is the first report on the application of multiomics, including transcriptome and metabolome analyses, to identify the putative genes for polysaccharide accumulation in *P. odoratum* and its relative species of the genus Polygonatum. In this study, the Q30 values of each sample were higher than those previously reported for *P. odoratum* (89.33%) and similar to those reported for *P. sibiricum* and *P. cyrtonema* (*Wang et al., 2017*; *Zhang et al., 2020*; *Li et al., 2022*). In addition, the number of raw reads for the samples tested in this study was higher than the transcriptome of *P. odoratum* reported in 2020 (*Zhang et al., 2020*). These results indicate that the quality of the transcriptome was higher and more reliable than that reported previously for *P. odoratum* (*Chen et al., 2014*). Meanwhile, a total of 14,194 DEGs and 80 DAMs were obtained by transcriptome and metabolome analysis (Fig. S2, Fig. 3), and these data laid a foundation for exploring and explaining the complex and changeable gene expression regulation mechanism and biological phenomena of polysaccharide accumulation in *P. odoratum.*

## Putative key DAMs involved in the polysaccharide accumulation in the rhizome of *P. odoratum*

In this study, we identified a total of 14,194 DEGs and 80 DAMs, of which 6,689 were up-regulated genes, and 52 were up-regulated in 'Y11' compared to 'Y10' (Fig. S2, Fig. 3). Based on the results of the KEGG enrichment analysis of the up-regulated DEGs and DAMs, we observed that seven up-regulated DAMs in 'Y11' were associated with carbohydrate metabolism (Fig. 4C). Among the seven up-regulated DAMs, the metabolite levan was a type of polysaccharide and constituted the major portion of polysaccharide (*Jiang et al., 2013*), suggesting that it is the major contributor for the polysaccharides accumulation in 'Y11'. Except for the metabolite levan, the two metabolites, glucose and beta-D-fructose 6-phosphate were involved in the proposed biosynthetic pathway of polysaccharides (Fig. 7), this suggests that the accumulation of polysaccharides is partly dependent on the biosynthesis of glucose and beta-D-fructose 6-phosphate (Figs. 3, 6). Another four metabolites that were not involved in the proposed biosynthetic pathway of polysaccharides, may function in the other biosynthetic pathway of the intermediate products of polysaccharides. For example, among these four metabolites, glucosaminic acid can be metabolically synthesised into glucosamine, which is phosphorylated intracellularly to synthesise glucose-6-phosphate, which can be further synthesised into fructose-6-phosphate *via* the plant sugar pathway (*Riegler et al., 2012*). As their important roles in the polysaccharide accumulation, these seven DAMs could be used as candidate quality markers for the assessment of medicinal quality of *P. odoratum.*

## Putative key DEGs acting in different manners to regulate polysaccharide accumulation in rhizome of *P. odoratum*

The pathways-genes-metabolites interaction network revealed that 17 DEGs could regulate the four putative key DAMs (Fig. 5). Among the 17 DEGs, the four DEGs (*HK, UGP2,*

*SUS* and *UGDH*) were involved in the proposed biosynthetic pathway of polysaccharides (Fig. 7). HK and ScrK are key enzymes in the biosynthesis of polysaccharides and can convert D-fructose into D-fructose 6-phosphate. Previous studies have reported that the activity of *scrK* regulates the sucrose metabolism in apple leaves (*Yang et al., 2018*), and HK is involved in modulating the sugar content in pears (*Zhao et al., 2019a*; *Zhao et al., 2019b*). In this study, beta-D-fructose 6-phosphate was observed to be upregulated in 'Y11' compared with that in 'Y10', two *scrK* DEGs and one *HK* DEGs were identified and showed higher transcript levels in 'Y11' than in 'Y10', and the expression level of *HK* was positively correlated with the beta-D-fructose 6-phosphate content (Fig. 5B). Coupled with the previous results that the expression of *scrK* was positively correlated with polysaccharide content in *P. sibiricum* and ginseng contents (*Wang et al., 2017*; *Fang et al., 2022*), and the expression levels of *HK* were positively correlated with the polysaccharide content in *P. cyrtonema* Hua (*Chen et al., 2022*), we speculated that *scrK* and *HK* genes were the possible key putative genes that positively regulate polysaccharide accumulation.

SUS was the key enzyme gene involved in the proposed biosynthetic pathway of polysaccharides. In this study, expression pattern analysis revealed that *SUS* gene preferably expressed in the rhizome, the main tissue for polysaccharides accumulating in *P. odoratum*, the transcript levels were opposite to the the trend of the polysaccharide content of the four cultivars, and was correlated with glucose, maltose and levan content (Figs. 5–6). In addition, the seven SNPs were found different in polysaccharide content in these four genotypes including two high polysaccharide content and two genotypes with low polysaccharide content (Fig. S5). Combined with the previous results that the expression levels of *SUS* were positively correlated with the polysaccharide and sucrose content and in *P. cyrtonema* Hua and ginseng contents (*Chen et al., 2022*; *Fang et al., 2022*), our results suggest that the *SUS* genes identified in this study may mediate the negative feedback regulation of the polysaccharide biosynthesis, and which also need to be further studied in the future.

## Transcriptional regulatory networks involved in polysaccharide accumulation in *P. odoratum*

Several studies have shown that different transcription factors play an important role in the regulation of polysaccharide accumulation (*Gao et al., 2021*; *Li et al., 2021*). For example, an endosperm-specific transcription factor, *TaNAC019*, regulates starch accumulation and improves the wheat grain quality (*Gao et al., 2021*). In apple, *MdWRKY32* is involved in starch-sugar metabolism by regulating the expression levels of *MdBam5* during post-harvest storage (*Li et al., 2021*). In the medicinal plant *P. cyrtonema Hua*, TFs such as *bHLH*, *bZIP*, *ERF* and *ARF* have been identified as potential regulators involved in polysaccharide accumulation (*Chen et al., 2022*). Similar with these findings, correlation analysis in this study revealed that *bHLH*, *ERF*, *NAC* and other TFs may control the polysaccharide accumulation in the rhizome of *P. odoratum* (Fig. 8 and Fig. S4). Previous studies suggested that TFs act as regulators in polysaccharide accumulation by regulating the key enzyme gene (*Gao et al., 2021*; *Li et al., 2021*; *Shi et al., 2022*), and future research will focus on the identification and functional analysis of target gene of these TFs.
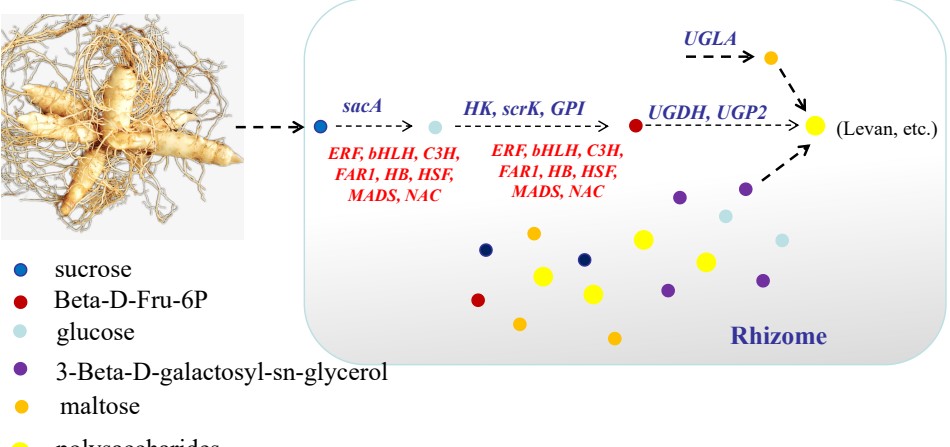

**Figure 10** **A model explaining the key genes and metabolites involved in polysaccharide accumulation in *P. odoratum*.** In the process of polysaccharide accumulation in the rhizome of *P. odoratum,* three metabolites including glucose, Beta-D-Fru-6P and levan were the key metabolites for polysaccharide accumulation. Among the these metabolites, glucose was regulated by the enzyme genes *sacA*, and beta-D-fru-6P was regulated by the genes *HK*, *GPI* and *scrK*. In addition, the transcription factors *C3H, FAR1, bHLH, ERF, HB, HSF, MADS* and *NAC* were also involved in regulating the accumulation of the two metabolites, glucose and beta-D-fru-6P. Meanwhile, two other key metabolites, maltose and 3-beta-D-galactosyl-sn-glycerol, involved in an unknown pathway to account for the accumulation of polysaccharide, and 3-beta-D-galactosyl-sn-glycerol accumulated in the rhizome may be associated with the expression of the gene *UGLA*.

## CONCLUSION

In conclusion, this study focused on the key genes and metabolites underlying polysaccharide accumulation in *P. odoratum* rhizomes using metabolomic and transcriptomic technologies. The integrated metabolomic and transcriptomic analyses revealed that DEGs and DAMs shared five pathways related to carbohydrate metabolism based on the KEGG enrichment analyses. Among these pathways, four DAMs and 17 DEGs were identified as key metabolites or key genes involved in the polysaccharide accumulation. The key enzyme genes including *sacA*, *HK*, *scrk* and *GPI,* may regulate the accumulation of two metabolites, glucose and Beta-D-Fru-6P, in the proposed polysaccharide biosynthetic pathway, which may be involved in the regulation of the transcription factors *C3H*, *FAR1*, *bHLH* and *ERF* (Fig. 10). Meanwhile, two other key metabolites, maltose and 3-Beta-D-galactosyl-sn-glycerol, involved in an unknown pathway to account for the accumulation of polysaccharide, and 3-Beta-D-galactosyl-sn-glycerol accumulated in the rhizome may be associated with the expression of the gene *UGLA* (Fig. 10). Although their biological functions in polysaccharide accumulation require further validation, this study is the first comprehensive analysis of polysaccharide accumulation using multiomics and lays a foundation for elucidating the molecular mechanisms of medicinal quality formation in *P. odoratum* rhizomes.

### Funding

This research was funded by the Natural Science Foundation of Hunan Province-China (2023JJ50053) and Hunan Traditional Chinese Medicine Research Project (B2023137). The funders had no role in study design, data collection and analysis, decision to publish, or preparation of the manuscript.

### Grant Disclosures

The following grant information was disclosed by the authors:
The Natural Science Foundation of Hunan Province-China: 2023JJ50053.
Hunan Traditional Chinese Medicine Research Project: B2023137.

### Competing Interests

The authors declare there are no competing interests.

### Author Contributions

- Gen Pan conceived and designed the experiments, performed the experiments, analyzed the data, prepared figures and/or tables, authored or reviewed drafts of the article, and approved the final draft.
- Jian Jin performed the experiments, prepared figures and/or tables, authored or reviewed drafts of the article, and approved the final draft.
- Hao Liu performed the experiments, analyzed the data, prepared figures and/or tables, and approved the final draft.
- Can Zhong analyzed the data, prepared figures and/or tables, and approved the final draft.
- Jing Xie performed the experiments, analyzed the data, prepared figures and/or tables, and approved the final draft.
- Yuhui Qin conceived and designed the experiments, performed the experiments, analyzed the data, authored or reviewed drafts of the article, and approved the final draft.
- Shuihan Zhang conceived and designed the experiments, performed the experiments, analyzed the data, authored or reviewed drafts of the article, and approved the final draft.

### Data Availability

The RNA-seq data of the rhizomes of the two materials is available at NCBI: PRJNA1084213.

The metabolomics data is available in NGDC: OMIX006030.

## Supplemental Information

Supplemental information for this article can be found online at http://dx.doi.org/10.7717/peerj.17699#supplemental-information.

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
