# Peer review of "Integrative analysis of the transcriptome and metabolome provides insights into polysaccharide accumulation in Polygonatum odoratum (Mill.) Druce rhizome"

_PeerJ, doi:10.7717/peerj.17699_

## Round 0.1 · original submission · Major Revisions

Revise article considering the suggestions of the reviewers

Reviewer 1 ·

Basic reporting

The article "Integrative analysis of the transcriptome and metabolome provides insights into polysaccharide accumulation in Polygonatum odoratum (Mill.) Druce rhizome" has professional article structure and is clear and unambiguous for the readers. The English language is sufficient along with the literature references.

Experimental design

The research question of the paper is well defined and fills an identified gap. Methods are described in detail and can be replicated.
Except for qRT-PCR, authors are encouraged to provide information on their housekeeping gene.

Validity of the findings

No comment

Additional comments

1) line 76 -79 could be rewritten as " Transcriptomic and metabolomic analyses have emerged as efficient methods for exploring complex biochemical pathways and identifying key genes and metabolites involved in the biosynthesis of bioactive compounds in plants"
2) to improve the Streamline sentences: line 44-47 could be rewritten as ""Polysaccharides, defined as polymer carbohydrates comprising at least ten monosaccharides linked by glycosidic bonds, play crucial roles in various physiological processes. In P. odoratum, they form a heteropolysaccharide composed of mannose, galactose, glucose, arabinose, and galacturonic acid, imparting diverse pharmacological properties such as blood glucose reduction, immune regulation, anti-tumor activity, antioxidant effects, anti-fatigue properties, and delayed skin aging (Deng et al., 2012; Zhao et al., 2019)."
3) line 152 what are the createria to choose DEG? Foldchnege? FDR? p-valus?
4) Please explain in the M&M how you identified the TFs.
5) Authors are requested to perform correlation analysis between DEGs and DEMs.
6) Rewrite the discussion and provide different sections with different subtitles to emphasise the importance of your findings.

Reviewer 2 ·

Basic reporting

The manuscript is interesting and highly useful for identification of gene responsible for polysaccharide accumulation in Polygonatum odoratum plants. The manuscript was well written.

Experimental design

The experimental design of the manuscript were scientifically sound and reliable.

Validity of the findings

The findings of the manuscript or DEGs in the Polysaccharide biosynthesis were validated through RT-PCR. However, additional works would be highly recommended for application in breeding programme or development of high polysaccharide content in this plant.
1. Expression analysis of DEGs in the rest of the 11 genotypes would be useful to further validate the role of these DEGs in polysaccharide accumulation.
2. identification of Sequence variation of major/important DEGs would be useful to develop functional markers for breeding programmme of polysaccharide content in Polygonatum odoratum plants.

---

## Round 0.2 · accepted · Accept

Reviewers have recommended the article for publication.

Reviewer 1 ·

Basic reporting

I have no further suggestions and believe that the manuscript is now suitable for publication.

Experimental design

I have no further suggestions and believe that the manuscript is now suitable for publication.

Validity of the findings

I have no further suggestions and believe that the manuscript is now suitable for publication.

Reviewer 2 ·

Basic reporting

The revised manuscript has substantial improvement and incorporated all the comments of the reviewers.

Experimental design

The experimental methods were well designed and found satisfactory.

Validity of the findings

The finding of the experiments were well validated in the manuscript.